# Clinical profile of alcohol dependent paintents according to Lesch Typology one year after the Covid-19 pandemic-comparative study

Dusan Kuljancic[1,2,¤a,¤b,☺,*], Jelena Amidzic[1,¤a,¤b,☺,*], Lazar Ljubotin[1,¤a,¤b,☺], Djendji Siladji[1,2,¤a,¤b,☺], Mina Cvjetkovic Bosnjak[1,2,¤a,¤b,☺], Vladimir Knezevic[1,2,¤a,¤b,☺], Dragana Ratkovic[1,2,¤a,¤b,☺], Vanja Bosic[1,2,¤a,¤b,‡], Vesna Vasic[1,2,¤a,¤b,‡], Branislav Sakic[1,2,¤a,¤b,‡], Sanja Bjelan[1,2,¤a,¤b,‡], Minja Abazovic[1,2,¤a,¤b,‡], Masa Comic[1,2,¤a,¤b,‡], Predrag Savic[1,2,¤a,¤b,‡]

**1** Department of Psychiatry and Clinical Psychology, Medical Faculty, University of Novi Sad, Novi Sad, Vojovodina, Serbia, **2** Clinics of Psichiatry, University Clinical Centre of Vojvodina, Novi Sad, Vojvodina, Serbia

☺ These authors contributed equally to this work.
‡ VB, VV, BS, SB, MA, MC, PS also contributed equally to this work.
¤a Current address: Department of Psychiatry and Clinical Psychology, Medical Faculty, University of Novi Sad, Novi Sad, Vojvodina, Serbia
¤b Current address: Clinics of Psichiatry, University Clinical Centre of Vojvodina, Novi Sad, Vojvodina, Serbia
* dusan.kuljancic@mf.uns.ac.rs (DK); 014841@mf.uns.ac.rs (JA)

## Abstract

### Background

The COVID-19 pandemics caused both physical and mental health problems raising global social tension, anxiety and discomfort which togheher lead to the increase in the consumption of psychoactive substances, among which alcohol was the most common, as a way of self-help. The hypothesis of this paper is the rising number of type II (anxious model) and type III (depressive model) alcohol dependent patients (as identified by the Lesch Typology) in the post-COVID-19 pandemic period compared to the pre-pandemic period, as a likely consequence of the stress, fear, problems and adversities that were caused by the pandemic.

### Method

The research was conducted as a retrospective cross-sectional study. It included 218 patients who were diagnosed with alcohol dependence. To classify the patients by the Lesch Typology, the MS Windows softer package for data processing available in public domain was used.

### Results

In relation to the Lesch Typology, 111 (50.9%) patients belonged to type III, 45 (20.6%) to type I, 37 (17.0%) to type II and 25 (11.5%) to type IV.

**Data availability statement:** There are ethical and legal restrictions on sharing a de-identified data set, because data contain potentially identifying or sensitive patient information taking into account that those are psychiatric patients with the addiction diagnosis. We are obligded to protect the privacy of the patients participants because this matter is so sensitive and easiy can geopardise the patients privacy and exposure them to the public wich could be disasterous and is not according to the law. The Ethical Commitee of Clinical Centre of Vojvodina obligded us to act like this. That is why we stated that the data will be shared upon the request to the corresponding authors. The request from the Ethical Commitee of our institution can be made on the phone No. +381214843484 or on the email address doktorspecijalistamedicine@gmail.com.

**Funding:** The author(s) received no specific funding for this work.

**Competing interests:** The authors have declared that no competing interests exist.

## Conclusion

Compared to the pre-pandemic findings of alcohol dependents classification according to the Lesch Typology, there was no increase in types II and III after the COVID-19 pandemic.

## 1. Introduction

The COVID-19 disease was caused by the new coronavirus SARS-CoV-2 (Severe Acute Respiratory Syndrome Coronavirus-2), discovered in late 2019 in China, where the infection first occurred. Soon after it developed its pandemic potential with more than 100 million people being infected and 2 million died worldwide. It ranges from asymptomatic to sever, even lethal multisystem forms [1,2]. However, in addition to physical, the COVID-19 pandemic also impacted people's mental health. It contributed to the increase of global social tension, anxiety and discomfort. There was an increase in the consumption of psychoactive substances, among which alcohol was the most common, as a way of self-help [3]. Killgore et al. have conducted a research to determine the relation between the lockdown and social isolation during the pandemic and the increased frequency of alcohol intake. They asked 5,931 individuals to complete the Alcohol Use Disorders Identification Test (AUDIT). Over a six-month period, the consumption of alcohol and likely dependence increased on a monthly basis for the individuals that were under stay-at-home restrictions compared to those who were not [4]. Due to the nature of their illness, psychiatric patients represent a vulnerable social group, and as such their reaction to the COVID-19 pandemic resulted in the worsening of their psychopathological symptoms, among which anxiety and depression were the most frequently recorded ones [3].

Treatment for The Alcohol Use disorder is a complex, long process which aims at establishing abstinence, behavioural changes and adopting healthy lifestyle habits. The success of treatment depends on the applied methods: with a conventional approach (pharmacotherapy and psychotherapy) the favourable outcome is achieved by one-fifth of patients, whereas with the systematic model by almost three-quarters of patients. However, there is no universal treatment plan which can be applied to all alcohol dependent patients. In order to plan a treatment, triage a patient and design an individual treatment plan, it is essential to have a systematised classification which can provide clear guidelines on different ways of treatment and optimal therapy goals [5].

The Lesch Typology is based on the data gathered from a longitudinal, prospective study. By observing The Alcohol Use disorder development and progression in 436 patients, 4 types of alcohol dependent patients were identified based on the patients' drinking patterns and origin of alcohol craving:

I.   the "allergy model" (craving caused by alcohol)

II.  the "anxiety model" (craving caused by stress)

III. the "depressive model" (craving caused by mood)

IV. the "conditioning model" (craving caused by compulsion).

Type I patients tend to use alcohol to reduce their withdrawal symptoms. If Type I patients abruptly decrease or end their usual alcohol intake, severe withdrawal symptoms, such as tremor, profuse sweating, restlessness and/or epileptic seizures arise. The withdrawal symptoms develop rapidly (often within hours) and disappear within a few days [6].

Type II patients when conflicts arise in their lives, they develop strong feelings of anxiety, where instead of dealing with their problems directly ("harm avoidance"), these patients rather turn to alcohol for its anxiolytic and calming effects. In such situations, alcohol is often abused as a conflict resolution strategy [6].

Type III patients abuse alcohol for its mood enhancing and sleep inducing properties. However, although alcohol seems to act soporifically, it actually destroys the sleep architecture, adding further to the patients' sleeping problems. Epileptic seizures in the patient's medical history are very rare, but should be considered. However, mood disorders are often found in the patients' family history [6].

During the phase of brain development (<14 years), and long before developing a drinking career, Lesch Type IV patients are often associated with significant childhood abnormalities. Traumatic brain injuries (with unconsciousness lasting longer than 6 hours) and cerebral diseases (e.g., meningitis) as well as deviant child behaviors (e.g., stuttering, nail-biting and/or bedwetting for a period longer than six months) can often be found in the patient's history. Tonic-clonic seizures, neither associated with alcohol consumption nor with alcohol withdrawal, are also frequently found in the patients' medical records. Type IV patients may also display gait disturbances caused by the severe polyneuropathy often seen in these patients [6].

The Lesch Typology has proven to be exceptionally useful in everyday clinical practice, as it offers both the classification of alcohol dependent patients and additional information on therapeutic goals, as well as the most appropriate treatment approach to each subtype of patients [6].

In 2017, hence before the COVID-19 pandemic, Vejnovic et al. conducted a survey of 105 alcohol dependents at the Clinic for Psychiatry, Clinical Centre of Vojvodina. The patients were classified in one of the four types based on the Lesch Typology. The majority of patients belonged to type III (60% of the respondents), then to type II (21.9%), type IV (10.5%) and to type I (7.6%) [7].

Our paper aims to compare the sociodemographic and clinical characteristics of the patients admitted to hospital for The Alcohol Use disorder treatment a year after the COVID-19 pandemic to the aforementioned results of the 2017 study.

The working hypothesis of this paper is the rising number of type II and type III alcohol dependent patients (as identified by the Lesch Typology) in the post-COVID-19 pandemic period compared to the pre-pandemic period, as a likely consequence of the stress, fear, problems and adversities that were caused by the pandemic, especially the change of routine, set patterns of living, fewer social contacts and constant uncertainty. Furthermore, we studied the protective effect of anti-COVID-19 vaccine on somatic complications in alcohol dependent patients and the length of their hospitalisation.

## 2. Materials and methods

The research was conducted as a retrospective cross-sectional study at the Addictions Unit of the Clinic for Psychiatry, Clinical Centre of Vojvodina in Novi Sad. It included 218 patients aged over 18 years who were diagnosed with alcohol dependence (ICD-10: F10.2) according to the ICD-10 criteria and who were admitted for treatment in the period from 3rd October 2022–31 October 2023, that is to say, after the last wave of the COVID-19 pandemic in Serbia in August 2022 [8]. All the patients included in the research gave their written consent to treatment and thus participation in this research when admitted to hospital. The patients with comorbidities, either psychiatric or somatic, were not excluded from the research. Furthermore, this is a comparative study as its findings are compared to the findings of a similar study carried out at the Clinic for Psychiatry 6 years earlier, that is, before the pandemic.

The data was collected by viewing and reviewing electronic medical records using the Clinical Information System (CIS). The data mentioned were accessed for research purposes from 1st December 2023 till 21st December 2023 and none of the authors had access to information that could identify individual participants during or after data collection. The

following sociodemographic characteristics were used: gender, age, marital status and occupation, while the clinical characteristics included: the time period of alcohol consumption, the number of hospitalisations due to TheAlcohol Use disorder, the duration of the last hospitalisation, lesions of the liver and nervous system, other psychiatric diagnoses, heredity to Alcohol Use disorder, whether the patient had a COVID-19 infection as well as their COVID-19 vaccination status.

To classify the patients by the Lesch Typology, the MS Windows LAT Instrument softer package for data processing available in public domain was used (www.lat-online.at). The Lesch alcoholism typology (LAT) was introduced in 1991. This instrument assesses items of major importance for the course and treatment of alcohol dependence, including age of onset, family history, co-morbidity, alcohol-related disabilities and the Lesch typology itself (in 11 questions to be filled in) [9]. These 11 items were then used to establish an instrument acting as a decision tree which, in the meantime, has been validated through basic and neurophysiologic research studies as well as in many treatment trials [6]. A computerized version of the LAT decision tree had been created and is being used on a daily basis in hospitals in many European countries. After completing the survey, the computer program determines the patient's typology, and additionally provides information about treatment goals and methods most appropriate for the patient's subtype [6]. The LAT Instrument has been validated in various countries, especially for pharmacological treatment of withdrawal and relapse prevention [9].

The obtained data was exported to Microsoft Excel and statistically processed. The IBM SPSS 27.0 software was used for descriptive and analytical processing of the data. The results were presented in tabular and graphical forms.

In order to determine the importance of vaccination against COVID-19 infection in the group of alcohol addicts, the length of the last hospitalization was used as the main characteristic. Vaccinated and non-vaccinated alcohol addicts were compared regarding the length of the last hospitalization. The length of hospitalization was taken as a marker because it is comprehensive and implies that there were no serious complications during treatment and that the course of treatment and the outcome were favorable. Given that this is an observational study, the outcome of the hospital treatment was not detailed, but it was generated and presented only by the length of the treatment.

The paper was written in accordance to the Declaration of Helsinki on medical research conduct. The Ethical Board of the Clinical Centre of Vojvodina gave their approval for the conduct of the research on 1st December 2023, registered under number 00–281.

## 3. Results

The research included 218 patients, of which 180 (82.6%) were male, and 38 (17.4%) were female (Fig 1).

The average age of the patients was 51 (SD ± 11) years. The youngest patient was 20, and the oldest was 77 years old. Based on their age, the patients were divided into several groups. The young adults group included patients ages 18–34, the group of the early middle-aged adults comprised patients ages 35–50, middle-aged patients were 51–65 years old,

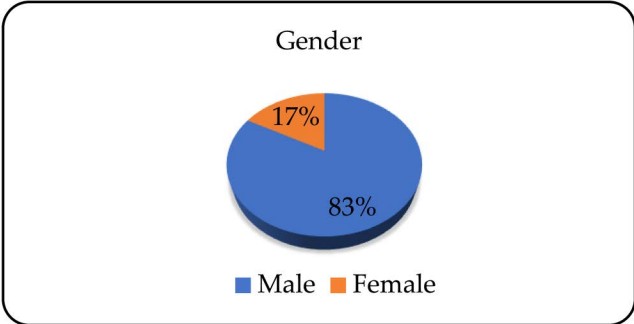

**Fig 1. Gender distribution of patients.**

and older adults over age 66. In our study, there were 87 (39.9%) middle-aged patients, 86 (39.5%) early middle-aged patients, 26 (11.93%) older adults and 19 (8.71%) young adults. The results are presented in a tabular form (Table 1).

Of the total number of patients, 79 (36.2%) were married, 75 (34.4%) were single, 58 (26.6%) were divorced, and 6 (2.8%) have lost their spouse/partner (Fig 2).

In relation to work status, 97 (44.5%) patients were unemployed, 94 (43.1%) were employed, and 27 (12.4%) were retired (Fig 3).

The length of the average period of using alcohol drinks was 20 (SD ± 11) years. The shortest period was 6 months, and the longest 50 years. The average number of hospitalisations of alcohol dependent persons was 2 (SD ± 1). The avreage duration of the last hospitalisation due to alcohol use disorder was 7 days (SD ± 5). The shortest hospitalisation lasted 1 day, and the longest 23 days.

Of the total number of patients, 161 (73.9%) did not have a positive family history of The Alcohol Use disorder, and 57 (26.1%) did (Fig 4).

In relation to the Lesch Typology, 111 (50.9%) patients belonged to type III, 45 (20.6%) to type I, 37 (17.0%) to type II and 25 (11.5%) to type IV. The results are presented in a tabular form (Table 2).

Of the total number of patients, 125 (57.3%) did not have liver damage, and 93 (42.7%) did (Fig 5).

**Table 1. Patients' age range.**

| Age group | Number of patients (n) | Percent (%) |
|---|---|---|
| Middle-aged adults (51-65) | 87 | 39.9 |
| Young middle-aged adults (35-50) | 86 | 39.5 |
| Older adults (≥66) | 26 | 11.93 |
| Young adults (18-34) | 19 | 8.71 |
| Total | 218 | 100 |

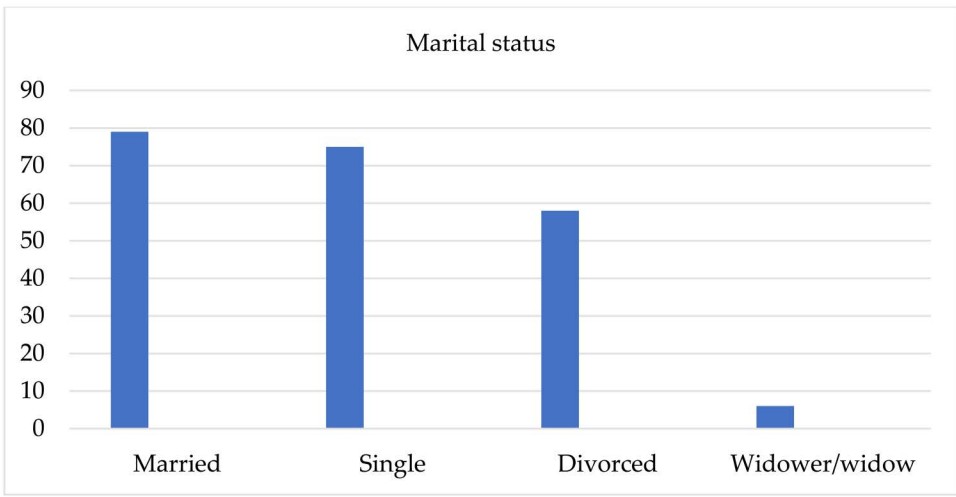

**Fig 2. Patients' marital status.**

Of the total number of patients, 138 (63.3%) did not have neurological damage, and 80 (36.7%) did (Fig 6).

In addition to alcohol dependence, 117 (53.7%) patients had a psychiatric comorbid condition and/or symptom, and 101 (46.3%) did not. Out of the total of 176 psychiatric comorbid conditions and symptoms, 69 (39.2%) suffered from depression, 31 (17.6%) suicidality, 28 (15.9%) personality disorder, 23 (13.1%) anxiety, 22 (12.5%) psychiatric disorders due to the use of psychoactive substances other than alcohol (heroin, marijuana, opiates, cocaine, speed and ecstasy) and 3 (1.7%) psychotic disorders. The results are presented in a tabular form (Table 3).

146 (66.9%) patients had a COVID-19 infection, of which 79 (36.2%) were treated on an ambulatory basis, and 62 (30.7%) in hospital conditions. The other 72 patients (33.1%) did not suffer from COVID-19 infection. The results are shown graphically (Fig 7).

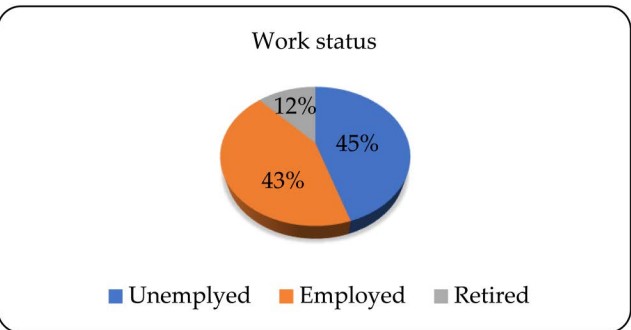

**Fig 3. Patients' work status.**

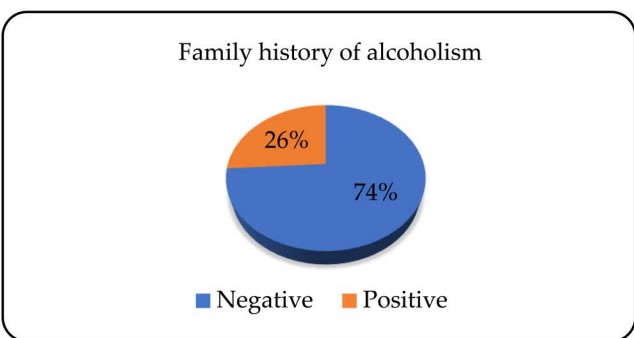

**Fig 4. Psychiatric heredity disposition to The Alcohol Use disorder.**

**Table 2. The Lesch Typology.**

| Type by the Lesch Typology | Nubmer of patients (n) | Percent (%) |
|---|---|---|
| III | 111 | 50.9 |
| I | 45 | 20.6 |
| II | 37 | 17.0 |
| IV | 25 | 11.5 |
| Total | 218 | 100 |

In relation to the vaccination status against COVID-19 infection, 112 (51.4%) patients received the vaccine, and 106 (48.6%) did not (Fig 8).

The Mann-Whitney test shows that there is a statistically significant difference in the average length of the last hospitalisation in relation to the COVID-19 vaccination status (p < 0.05).

## 4. Discussion

Soon after its initial outbreak in China, the COVID-19 disease was declared a global pandemic by the WHO in March 2020 [10]. Alongside its great impact on people's physical health, social functioning and economics, this public health issue has

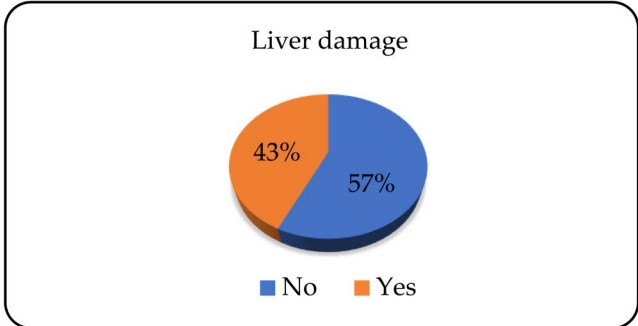

**Fig 5. Presence of liver damage.**

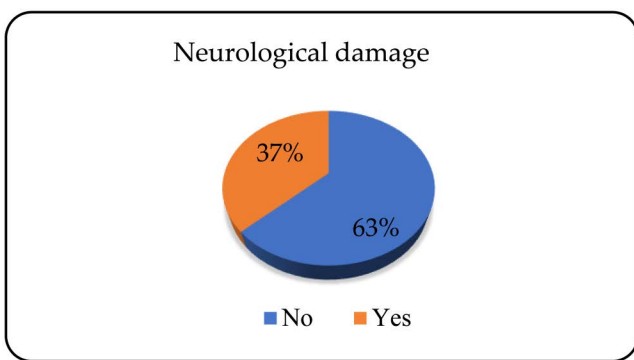

**Fig 6. Presence of neurological damage.**

**Table 3. Psychiatric comorbid conditions and symptoms.**

| Psychiatric diagnosis | Number (n) | Percent (%) |
|---|---|---|
| Depression | 69 | 39.2 |
| Suicidality | 31 | 17.6 |
| Personality disorder | 28 | 15.9 |
| Anxeity | 23 | 13.1 |
| Disorders due to the use of PS other than alcohol | 22 | 12.5 |
| Psychotic disorders | 3 | 1.7 |
| Total | 176 | 100 |

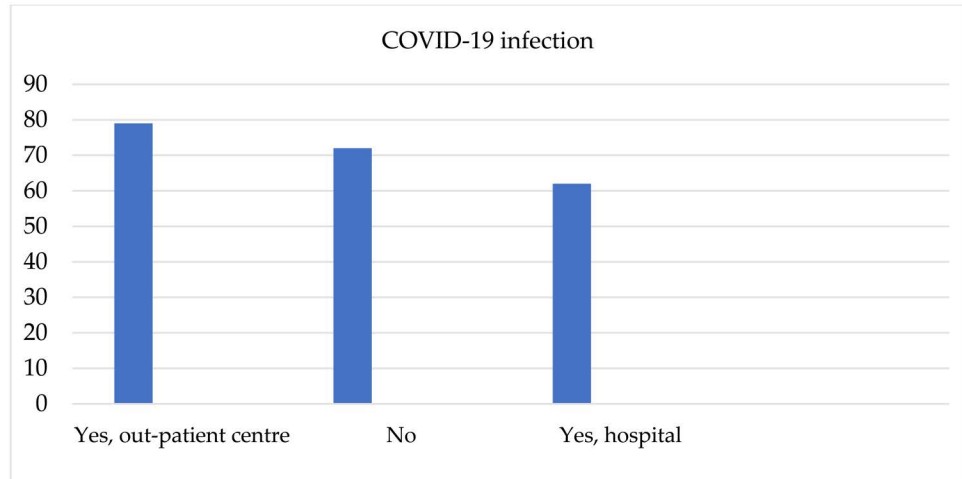

**Fig 7. Presence of COVID-19 infection.**

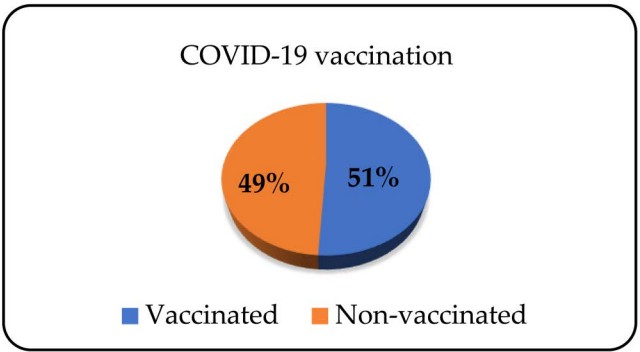

**Fig 8. Patients' status of vaccination against COVID-19 infection.**

a significant effect on their mental wellbeing, as well as on the life quality of each individual [11]. Restrictive measures such as isolation and quarantine, aimed at suppressing this contagious disease are the key to preventing the transmission of the virus, but they represent a psychological burden for a population, disrupting personal, family and social functioning. The consequences of the pandemic are significant among vulnerable groups such as psychiatric patients, who are on the social margin even in normal circumstances [12].

The change in day-to-day routine and set patterns of living imposed by the pandemic caused a sense of fear and anxiety among vulnerable individuals, thus increasing incidences of stress-related disorders, but also causing exacerbation of existing psychiatric disorders. Strict epidemiological control measures, acting as social stressors, may lead to serious mental disorders such as depression and/or anxiety among previously healthy individuals or represent a huge burden for the mentally ill [12]. The appearance of fear, anxiety and depressive symptoms heightens the desire to consume psycho-active substances such as alcohol, as a form of self-help, especially during stressful periods [13].

In this study, a statistically significantly higher presence of alcohol dependence was observed in males. In the United States of America (USA), alcohol use disorder diagnosed more often in men (7%) than in women (4%) [14]. The incidence of alcohol dependence during the pandemic was higher among males in Canada, India, Iran and the USA [15]. The

average age of patients dependent on alcohol was 51 years, with almost equal representation of young middle-aged and middle aged adults. The National Epidemiologic Survey on Alcohol and Related Conditions (NESARC) and the National Survey on Drug Use and Health (NSDUH) have similar results related to the age ranges of persons dependent on alcohol. The prevalence of alcohol use disorders decreases with age. The risk of developing alcohol dependence was higher among the respondents aged 30–54 years compared to those who were 55 years or older [16]. White et al. pointed to the increase in number of alcohol-related deaths in 2020 compared to 2019, with the highest mortality rate in the age group 55–64 years [17].

In relation to marital status, the majority of patients are married, a slightly smaller number are single, while a quarter are divorced. Research conducted in June 2021 showed that there was a lower chance of getting married, that is, a higher chance of divorce among the population of alcohol dependents [18]. In relation to their work status, the largest number of patients are unemployed, a slightly smaller number are employed, and the fewest are retired. In their research on the use of psychoactive substances in Israel during the COVID-19 pandemic, Levy and colleagues concluded that the differences in work status were significant only among the respondents who used illegal drugs but not alcohol. The occurrence of equal representation of the employed and unemployed among the alcohol dependent persons is explained by the wide availability, popularity and social acceptability of alcohol beverages in Israel, which is also the case in Serbia [19].

On average, the number of hospitalisations of alcohol dependents is 2, and the average duration of the last hospitalisation is 7 days. Ahacic et al. have conducted a study which included 576 patients admitted to hospital for alcohol dependence. Nearly 50% of the patients had been hospitalised only once before, 34% three times or more, and 18% twice. Most patients were readmitted for treatment in the period between 2 and 4 years after the last hospitalisation [20]. Our study has found that almost three quarters of patients have a negative family history of alcohol dependence. Edenberg and colleagues point to the complexity of alcohol dependence as a genetic disease with variations in a large number of genes which increase the risk of the disease. Among the identified genes, the ones that stand out are ADH1B and ALDH2, which are included in the metabolism of alcohol and have the greatest impact on the manifestation of alcohol dependence. Twin studies in the USA and Europe suggest that the concordance for alcohol dependence ranges between 45% and 65% [21].

In relation to the Lesch Typology, 51% of the patients belonged to Type III, 21% to Type I, 17% to Type II and 11% to Type IV. In the period between 2006 and 2007, Penha Zago-Gomes and Nakamura-Palacios have conducted research on alcohol dependent patients at the Universidade Federal do Espírito Santo hospital. Of 170 patients, 21.2% were classified as Type I, 29.4% as Type II, 28.8% as Type III, and 20.6% as Type IV. They concluded that, although different types of alcohol addicts have different clinical presentations, the age at first intake of alcohol and the length of withdrawal did not differ among the types according to Lesch's Typology [22]. Jakovljevic and colleagues compared clinical characteristics of alcohol dependents in Serbia and Austria in the period between 2011 and 2012. The sample of Serbian population showed a very high rate of anxiety disorders (89.8% compared to 26.5%), which was explained by traumatic historical events in Serbs. Suicidal tendencies, not related to alcohol use or withdrawal syndrome, are significantly higher in the Austrian population sample (1.6% compared to 13.2%). There was no difference between the two samples in Type IV patients (20.5% compared to 20.6%). The sample of our population had significantly more Type III patients (42.5% compared to 33.8%) [23]. When compared to the results of classification of alcohol dependents according to the Lesch Typology obtained by Vejnovic et al. in 2017, it is evident that there was no increase in type II or type III after the COVID-19 pandemic, hence we reject the working hypothesis. Almost the same number of Type IV and a slightly higher number of Type I dependents is observed in our study. A study on reaction of the population of 4 different countries (Lebanon, Portugal, Italy and Serbia) to the COVID-19 pandemic indicated that the respondents from Serbia had greater resistance to psychological difficulties and stress that the COVID-19 pandemic imposed with its epidemiological measures than the other participating countries. The authors explain that the reason for this lies in specific sociocultural characteristics of Serbian population, such as previous experience with epidemics and the wars in Yugoslavia in the period between 1991 and 1999 [24].

Our hypothesis was not confirmed. This is largely due to the peculiarity of our population, which has suffered numerous stressful national tragic events in the last 30 years. However, on the other hand, many experts around the world point out that the cumulative effects of stress and the consequences of epidemiological measures related to the COVID19 pandemic will be felt in the coming period and will very likely long outlast the COVID19 pandemic even up to 3 years later [25,26].

Therefore, perhaps the examined period that we took in this research is still insufficient to record the increase in the anxious and depressive type of alcohol consumption patern. The examined period itself and the expected prolonged effect of the pandemic on mental health may be one of the explanations for the refutation of our hypothesis in this research.

However, as the problem of alcohol addiction is complex and polyetiologically conditioned, abovementioned explanation may also be insufficient. It cannot be forgotten that the Republic of Serbia and the entire health system here, in addition to engaging in the treatment and care of Covid patients, still promoted networks to help mentally unstable persons, vulnerable groups in this global public health crisis. There were SOS lines for psychological help and support, counseling centers for Alcohol Use disorder, mental health clubs that offered help to patients, resolution of current reactive feelings such as anxiety and depression, which, in addition to the proven cultural resistance to stress, largely contributed to the mitigation and reduction of perceived levels of anxiety and depression in vulnerable individuals, which resulted in a reduction in the use of alcohol as a form of self-medication and explains our results in this research [27].

We explain the increase of Type I, somatic and neurological comorbidities by the circumstances of the pandemic which led to difficulties in receiving medical assistance, non- treatment or inadequate treatment of non-COVID-19 diseases and conditions, given that the entire healthcare system was focused on the current pandemic. The COVID-19 pandemic has made significant disruption in health service delivery all around the world [28]. The unavailability of health care for non-COVID patients during the COVID-19 pandemic was caused both by the burden on the health system by a large number of COVID patients and by the overload and saturation of health workers who were mostly reassigned from their regular tasks to work in the COVID centers [28]. Takinig all mentioned together, it is clear why the COVID-19 pandemics has disrupted both preventive and curative services for communicable and noncommunicable diseases [29–31]. Many of essential services, like internal and neurological check-ups, diagnostic procedures, have been delayed by the healthcare facilities, [32–34] patients were also unable to attend follow-ups and acute care visits due to the fear and anxiety they experienced during the pandemic waves [35].

It was difficult for all categories of non-COVID patients to obtain adequate health care during the pandemic, and especially for vulnerable social groups such as the alcohol addicts. The pandemic itself and the changes caused by it in the health system led to an increase in tension and uncertainty among the alcohol addicts, which contributed to a change in the way of drinking alcohol as a coping mechanisam, which usually ment an increase in the amount and frequency of drinking, and easily led to the appearance of alcohol-induced disorders such as delirium tremens or abstinence syndrome, which led to acute hospitalization [36]. These disorders are a main feature of the first type of alcohol addicts according to the Lesh typology. The restricted access to healthcare system during the pandemic may have influenced the patterns of alcohol dependence leading to the increase in somatic complication of alcocholisam and explaining our finding- the increase of Lesh Type I addicts in our sample. This discovery sheds new light on how external stressors, such as the unavailability of adequate health care, can lead to a change in the pattern of alcohol consumption.

Nearly 60% of patients have no lesions of the liver, that is, there is no elevation of the liver enzymes. In the study conducted in Kentucky, 69% of alcohol dependents admitted for treatment had elevated biomarkers (aspartate amino-transferase and cytokeratin 18) for liver damage, with women having more pronounced lesions compared to men [37]. Neurological damage was present in 37% of patients. Meta-analysis results from 2018 indicated a prevalence of 46% of alcohol dependents with peripheral neuropathy [38].

In addition to alcohol dependence, most patients suffered from a psychiatric comorbid condition or symptom. Depression was the most prevalent with nearly 40%, followed by suicidality, personality disorder, anxiety and the use of psycho-active substances other than alcohol with almost the same share, and there were only three patients with a psychotic

disorder. Comorbid psychiatric disorders in patients with alcohol dependence are very common. On the one hand, alcohol dependence can induce the appearance of psychiatric disorders and on the other, disorders such as depression and anxiety can cause excessive intake of alcohol beverages as a form of self-medication [39]. Depressive disorders are most common among alcohol dependents, especially major depressive disorder with a prevalence of 33% as found in a 2019 study [40].

67% of patients were infected with COVID-19. The number of patients treated in an outpatient setting was almost equal to the number of those who were treated in hospitals. 52% of patients received a vaccine against COVID-19. Bailey et al. compared the course of the COVID-19 disease among patients with and without alcohol dependence. They proved that, when infected with SARS-CoV-2 virus, alcohol dependant patients have a higher chance of being admitted to hospital compared to patients without alcohol dependence. Moreover, the course of the disease is more sever, the complications are more prominent and mortality rates higher [41].

Literature data indicate the fact that patients with psychiatric disorders fall into a vulnerable category and if infected with the COVID-19 virus, they suffer more serious forms of the disease. In the population of psychiatric patients, the highest vaccination rate was among those with depression and anxiety disorders. On the other hand, patients suffering from psychotic disorders had the lowest rate of vaccination. There are several possible explanations as to why unvaccinated psychiatric patients had more severe forms of the COVID-19 infection. It is thought that they often ignored somatic symptoms and had less contact with primary health care. Also, it is possible that doctors focused more on the mental problems of these patients, and neglected somatic difficulties [42]. Findings from our study indicated a statistically significant difference in the average duration of the last hospitalisation compared to the vaccination status against COVID-19. The fully vaccinated patients were hospitalised for The Alcohol Use disorder for a shorter period and the course of the treatment was without serious complications. Studies on COVID-19 vaccination in the period between January and May 2022 have shown high efficacy of all available vaccines in the period [43]. The vaccination led to a smaller number of critically ill patients requiring treatment in intensive care units and a lower mortality rate [44].

There are some deficiencies and possible limitations of this study. First, the design of the research, it self, as an observational study of a retrospective nature does not provide the ability to determin the causality between the examined phenomena. To overcome this problem and to better understand casual relationships, a longitudinal study is needed to validate our results and make them generalizable. Second, the sample size may not be sufficient to account for the sample representativness. It is necessary to conduct a similar study in cooperation with several different psychiatric institutions throughout Serbia and region to shed a new light on the the complexities of alcohol abuse during crises and draw validated conclusions.

## 5. Conclusions

Compared to the pre-pandemic findings of alcohol dependents classification according to the Lesch Typology, there was no increase in types II and III after the COVID-19 pandemic. This might be explained by specific sociocultural characteristics of Serbian population and their higher tolerance of difficulties and stress imposed by the pandemic. However, we explain that the rise of type I alcohol dependents, with somatic and neurological comorbidities, was caused by the circumstances of the pandemic that led to difficulties in healthcare delivery and poor treatment of non-COVID related illnesses and the reorientation of the healthcare system to the pandemic treatment. More than a half of patients in our study had COVID-19, half of whom had been vaccinated. We have proven a positive effect of vaccination on the length and course of hospitalisation, which indicates the importance of implementing a full and comprehensive vaccination programme for vulnerable psychiatric patients, who are often neglected due to stigma.

**Informed consent statement**

Written informed consent was obtained from all subjects involved in the study.

## Acknowledgments

On this occasion, the authors would like to wholeheartedly thank the employees of the Psychiatry Clinic of the University Clinical Center of Vojvodina in Novi Sad for their technical, moral and advisory support during the preparation of this work, as well as for the time and space allocated for the collection of data used in the preparation of this study.

## Author contributions

**Conceptualization:** Dusan Kuljancic, Jelena Amidzic.

**Data curation:** Lazar Ljubotin, Djendji Siladji.

**Formal analysis:** Dusan Kuljancic, Vesna Vasic, Masa Comic.

**Investigation:** Vladimir Knezevic, Vanja Bosic.

**Methodology:** Jelena Amidzic, Branislav Sakic, Minja Abazovic.

**Project administration:** Dragana Ratkovic, Sanja Bjelan.

**Resources:** Djendji Siladji, Dragana Ratkovic.

**Software:** Vanja Bosic, Sanja Bjelan.

**Supervision:** Vladimir Knezevic, Branislav Sakic.

**Validation:** Mina Cvjetkovic Bosnjak, Vesna Vasic.

**Visualization:** Lazar Ljubotin, Predrag Savic.

**Writing – original draft:** Mina Cvjetkovic Bosnjak, Masa Comic.

**Writing – review & editing:** Minja Abazovic, Predrag Savic.

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
