## [Decision Letter · Decision Letter 0]

4 Oct 2024

PONE-D-24-35321Clinical Profile Of Alcohol Dependent Paintents According To Lesch Typology One Year After The Covid-19 Pandemic-Comparative StudyPLOS ONE

Dear Dr. Kuljancic,

Thank you for submitting your manuscript to PLOS ONE. After careful consideration, we feel that it has merit but does not fully meet PLOS ONE’s publication criteria as it currently stands. Therefore, we invite you to submit a revised version of the manuscript that addresses the points raised during the review process. 

We look forward to receiving your revised manuscript.

Kind regards,

Souparno Mitra, M.D.

Academic Editor

PLOS ONE

Journal Requirements:

**Additional Editor Comments:**

**Please address Reviewer 1s comments. Once these comments are addressed, the publication will be further considered for publication. **

Reviewers' comments:

Reviewer's Responses to Questions

**Comments to the Author**

1. Is the manuscript technically sound, and do the data support the conclusions?

Reviewer #1: Yes

Reviewer #2: Yes

2. Has the statistical analysis been performed appropriately and rigorously? 

Reviewer #1: Yes

Reviewer #2: Yes

3. Have the authors made all data underlying the findings in their manuscript fully available?

Reviewer #1: Yes

Reviewer #2: Yes

4. Is the manuscript presented in an intelligible fashion and written in standard English?

Reviewer #1: Yes

Reviewer #2: Yes

5. Review Comments to the Author

Reviewer #1: First, I’d like to commend the authors on producing a well-organized and timely manuscript. The study presents an important investigation into the clinical profiles of alcohol-dependent patients one year post-COVID-19, based on the Lesch Typology. The data-driven rejection of the hypothesis regarding an increase in Type II and III alcohol dependence is particularly valuable, and I believe this study makes a meaningful contribution to our understanding of how the pandemic has affected alcohol use patterns.

That being said, I’d like to offer a few suggestions for improvement, which I believe could strengthen the paper even further.

1. Expanding on the Increase in Type I Dependence:

While the manuscript does touch on the observed rise in Type I dependence, I think this aspect of the findings deserves more attention. The increase could be tied to the disruptions in healthcare services during the pandemic, which may have left some individuals unable to receive timely care for their somatic conditions. As a result, alcohol might have been used as a coping mechanism. Delving into how these healthcare challenges could have contributed to this rise would add valuable context to the findings.

Suggestion: It might be worth adding a section to the discussion focused on “Pandemic Healthcare Disruptions and the Rise of Type I Dependence.” By exploring how restricted access to healthcare during the pandemic may have influenced the patterns of alcohol dependence, the manuscript could offer a deeper understanding of this shift. This would also tie in nicely with the broader implications of how external stressors, like healthcare availability, can impact addiction patterns.

2. Clarifying the Impact of Vaccination on Hospitalization Outcomes:

The manuscript highlights the positive effects of COVID-19 vaccination on hospitalization outcomes, which is a key finding. However, I feel that the methodology for how this effect was measured could be explained in more detail. Were specific health markers or hospitalization duration compared between vaccinated and non-vaccinated patients? Were there control variables to account for differences in health status or severity of alcohol dependence?

Suggestion: Providing more details about the statistical methods used to assess the impact of vaccination would really help clarify these findings. For example, how were hospitalization outcomes specifically measured in vaccinated versus non-vaccinated patients? This could strengthen the conclusions around vaccination’s benefits and support stronger recommendations for its role in managing vulnerable populations like alcohol-dependent individuals.

3. Addressing Limitations and Future Research:

I appreciate that the manuscript acknowledges some limitations, but I believe expanding on this could further enhance the paper. For instance, the retrospective nature of the study poses certain challenges, and this could be discussed in more depth. Additionally, proposing future research directions—such as longitudinal studies or multi-center collaborations—would offer a proactive way to build on this study’s findings.

Suggestion: Including a more detailed "Limitations and Future Directions" section would be beneficial. By suggesting prospective studies that track alcohol dependence patterns over time, or studies that investigate these trends across different regions, the authors could provide a roadmap for future research. This would not only address the limitations but also showcase how this study sets the stage for ongoing inquiry in the field.

Conclusion:

I believe this is a well-executed and valuable study that addresses an important issue. By expanding on the rise in Type I dependence, clarifying the methodology behind the vaccination findings, and providing more insight into the study's limitations and future directions, the manuscript could be further strengthened. I genuinely look forward to seeing how these revisions will enhance the final version of the paper.

Thank you again for the opportunity to review this insightful and relevant work.

Reviewer #2: While this article effectively demonstrates the psychosocial aspects of addiction to psychoactive substances, specifically alcohol, it also provides a comprehensive analysis of the various factors involved in alcohol abuse. The well-written article effectively delves into the psychological and social elements contributing to alcohol addiction. However, it should be noted that the article does have limitations, including potential selection bias, lack of generalizability, and some degree of sample bias. Despite these flaws, the article still offers valuable insights into the complexities of alcohol abuse.

6. PLOS authors have the option to publish the peer review history of their article (what does this mean? ). If published, this will include your full peer review and any attached files.

**Do you want your identity to be public for this peer review?** For information about this choice, including consent withdrawal, please see our Privacy Policy .

Reviewer #1: No

Reviewer #2: **Yes: ** Rajasekhar Kannali

---

## [Author Response · Author response to Decision Letter 1]

20 Nov 2024

Dear Academic Editor and Reviewers,

First of all, I would like to express my gratitude for your positive assessment of my scientific work. We are aware that there are certain shortcomings in our article and we are grateful for your comments, which we hope have significantly contributed to improving the quality and scientific value of our article.

I will now present our responses to your objections and suggestions. I hope we were able to respond to your requests and thereby improve the quality of the work enough to be eligible for publication.

Respons to the reviewer 1 comments:

1. A section to the discussion focused on “Pandemic Healthcare Disruptions and the Rise of Type I Alcochol Dependence” has been added to the discusion chapter and the matter has been extesively explained citing new literature sources. I hope we met your request.

2. The methodology for how the effect of the Impact of Vaccination on Hospitalization Outcomes in our sample was measured has been explained in more detail in the Metodology section. I hope you will find it satissfactory.

3. A more detailed "Limitations and Future Directions" section has been added to the conclusion addresing both reviewer՚s comments. I hope it will be enough.

Respons to The Journal Requirements:

1. I hope my revised manuscript now meets the PLOS ONE's style requirements, including those for file naming. I included the tamplates you have suggested.

2. There are ethical and legal restrictions on sharing a de-identified data set, because data contain potentially identifying or sensitive patient information taking into account that those are psychiatric patients with the addiction diagnosis. We are oblidged to protect the privacy of the patients participants because this matter is so sensitive and easiy can geopardise the patients privacy and exposure them to the public wich could be disasterous and is not according to the law. The Ethical Commitee of Clinical Centre of Vojvodina oblidged us to act like this. That is why we stated that the data will be shared upon the request to the corresponding authors. The request from the Ethical Commitee of our institution can be made on the phone No. +381214843484.

3. We stated in the availability form like this according to ourՙs Ethical commitee decision on our data sharing protocol. The data can be shared only on request to the corresponding authors.

4. The ethics statement appear only in the Methods section of our manuscript.

5. A separate caption for each figure has been added in the manuscript.

6. Review of the reference list has been done to ensure that it is complete and correct.

Dear Editor and reviewers, I hope that with these minimal changes that we have made in our scientific work and according to your requests, it has now been improved and meets the criteria for publication.

I hope for a soon and positive answer.

Sincerely,

Dr. Dusan Kuljancic

---

## [Decision Letter · Decision Letter 1]

10 Jan 2025

PONE-D-24-35321R1Clinical Profile Of Alcohol Dependent Paintents According To Lesch Typology One Year After The Covid-19 Pandemic-Comparative StudyPLOS ONE

Dear Dr. Kuljancic,

Thank you for submitting your manuscript to PLOS ONE. After careful consideration, we feel that it has merit but does not fully meet PLOS ONE’s publication criteria as it currently stands. Therefore, we invite you to submit a revised version of the manuscript that addresses the points raised during the review process.

We look forward to receiving your revised manuscript.

Kind regards,

Souparno Mitra, M.D.

Academic Editor

PLOS ONE

Journal Requirements:

**Additional Editor Comments:**

Please address the reviewers comments in order to consider for acceptance. 

Reviewers' comments:

Reviewer's Responses to Questions

**Comments to the Author**

1. If the authors have adequately addressed your comments raised in a previous round of review and you feel that this manuscript is now acceptable for publication, you may indicate that here to bypass the “Comments to the Author” section, enter your conflict of interest statement in the “Confidential to Editor” section, and submit your "Accept" recommendation.

Reviewer #3: (No Response)

Reviewer #4: All comments have been addressed

2. Is the manuscript technically sound, and do the data support the conclusions?

Reviewer #3: Yes

Reviewer #4: Yes

3. Has the statistical analysis been performed appropriately and rigorously? 

Reviewer #3: I Don't Know

Reviewer #4: I Don't Know

4. Have the authors made all data underlying the findings in their manuscript fully available?

Reviewer #3: Yes

Reviewer #4: Yes

5. Is the manuscript presented in an intelligible fashion and written in standard English?

Reviewer #3: Yes

Reviewer #4: Yes

6. Review Comments to the Author

Reviewer #3: In the introduction -Please go into more detail regarding the exact proposed definition of each type. so that the readers have a thorough understanding of each type.

Please avoid using terms such as "Alcoholic" and "Alcoholism" - Instead use "Alcohol Use disorder".

See the remaining revierw comments in the attached PDF.

Reviewer #4: The previous suggestions have been incorporated. I have a couple more recommendations for a minor revision before being accepted.

1. The hypothesis regarding an increase in type II and III alcohol dependence was not supported by the data, yet the study does not explore alternative explanations sufficiently. Delve deeper into sociocultural and systemic factors that may explain the stability in type II and III dependency, such as support networks or societal resilience.

2. Provide more information about the software used for Lesch typology, including validation and reliability metrics.

3. Conduct thorough proofreading and use professional editing services to improve clarity and readability.

7. PLOS authors have the option to publish the peer review history of their article (what does this mean? ). If published, this will include your full peer review and any attached files.

**Do you want your identity to be public for this peer review?** For information about this choice, including consent withdrawal, please see our Privacy Policy .

Reviewer #3: **Yes: ** Arun Prasad

Reviewer #4: **Yes: ** Nikhil Tondehal

---

## [Author Response · Author response to Decision Letter 2]

9 Feb 2025

Dear Academic Editor and Reviewers,

First of all, I would like to express my gratitude for your positive assessment of my scientific work. We are aware that there are certain shortcomings in our article and we are grateful for your new comments, which we hope have significantly contributed to improving the quality and scientific value of our article.

I will now present our responses to your objections and suggestions. I hope we were able to respond to your requests and thereby improve the quality of the work enough to be eligible for publication.

Respons to the reviewer 1 comments:

1. The Introduction section has been expanded with more detail on each one of the 4 Lesch types of alcohol addiction.

2. Furthermore, the the terminology you requested is changed throughout the text.

Respons to the reviewer 2 comments:

1. As for the explanation for the rejection of our hypothesis, it was indeed unsatisfactory. We have now offered several alternative explanations that together can explain the discovery we made.

2. All information we could find online about the Lesch typology software we used was included in the study. Other data on the metric characteristics of the software are not publicly available, at least to us. I hope this information we have provided will be satisfactory.

3. Regarding the quality of the language and the readability and comprehensibility of the text, we did our best. Hopefully the text is of better quality now.

Dear Editor and reviewers, I hope that with these minimal changes that we have made in our scientific work and according to your requests, will now be enough to improve sufficiently the manuscript and meet the criteria for publication.

I hope for a soon and positive answer.

Sincerely,

Dr. Dusan Kuljancic

---

## [Decision Letter · Decision Letter 2]

2 May 2025

Clinical Profile Of Alcohol Dependent Paintents According To Lesch Typology One Year After The Covid-19 Pandemic-Comparative Study

PONE-D-24-35321R2

Dear Dr. Kuljancic,

We’re pleased to inform you that your manuscript has been judged scientifically suitable for publication and will be formally accepted for publication once it meets all outstanding technical requirements.

Kind regards,

Souparno Mitra, M.D.

Academic Editor

PLOS ONE

Additional Editor Comments (optional):

Reviewers' comments:

Reviewer's Responses to Questions

**Comments to the Author**

1. If the authors have adequately addressed your comments raised in a previous round of review and you feel that this manuscript is now acceptable for publication, you may indicate that here to bypass the “Comments to the Author” section, enter your conflict of interest statement in the “Confidential to Editor” section, and submit your "Accept" recommendation.

Reviewer #2: All comments have been addressed

Reviewer #3: All comments have been addressed

2. Is the manuscript technically sound, and do the data support the conclusions?

Reviewer #2: Yes

Reviewer #3: Yes

3. Has the statistical analysis been performed appropriately and rigorously? 

Reviewer #2: Yes

Reviewer #3: Yes

4. Have the authors made all data underlying the findings in their manuscript fully available?

Reviewer #2: Yes

Reviewer #3: Yes

5. Is the manuscript presented in an intelligible fashion and written in standard English?

Reviewer #2: Yes

Reviewer #3: Yes

6. Review Comments to the Author

Reviewer #2: This study examined issues related to substance abuse and addiction in the post-pandemic context. The study design is appropriate; however, it did not confirm the anticipated increase in alcohol consumption driven by anxiety. Instead, it highlighted how pandemic-related stress has affected drinking patterns. The sample size is small, which limits generalizability and makes it difficult to account for confounding factors. Additionally, the strain on the healthcare system and limited access to medical services have influenced trends in alcohol dependence. Future research is essential to assess the long-term psychological effects of the pandemic on alcohol use.

Reviewer #3: Thanks for the edits to the article. Hopefully we'll have more data in the future on the same which will strengthen the study.

7. PLOS authors have the option to publish the peer review history of their article (what does this mean? ). If published, this will include your full peer review and any attached files.

**Do you want your identity to be public for this peer review?** For information about this choice, including consent withdrawal, please see our Privacy Policy .

Reviewer #2: No

Reviewer #3: **Yes: ** Arun Prasad

---

## [Editor Report · Acceptance letter]

PONE-D-24-35321R2

PLOS ONE

Dear Dr. Kuljancic,

I'm pleased to inform you that your manuscript has been deemed suitable for publication in PLOS ONE. Congratulations! Your manuscript is now being handed over to our production team.

Kind regards,

on behalf of

Dr. Souparno Mitra

Academic Editor

PLOS ONE